# The Impact of Meal Dietary Inflammatory Index on Exercise-Induced Changes in Airway Inflammation in Adults with Asthma

**DOI:** 10.3390/nu14204392

**Published:** 2022-10-19

**Authors:** Katrina P. McDiarmid, Lisa G. Wood, John W. Upham, Lesley K. MacDonald-Wicks, Nitin Shivappa, James R. Hebert, Hayley A. Scott

**Affiliations:** 1Nutrition and Dietetics, School of Health Sciences, University of Newcastle, Callaghan, NSW 2308, Australia; 2Immune Health Research Program, Hunter Medical Research Institute, Lot 1 Kookaburra Circuit, New Lambton Heights, NSW 2305, Australia; 3School of Biomedical Sciences and Pharmacy, University of Newcastle, University Drive, Callaghan, NSW 2308, Australia; 4Lung and Allergy Research Centre, Diamantina Institute, The University of Queensland, Brisbane, QLD 4102, Australia; 5Food and Nutrition Research Program, Hunter Medical Research Institute, Lot 1 Kookaburra Circuit, New Lambton Heights, NSW 2305, Australia; 6Cancer Prevention and Control Program, Department of Epidemiology and Biostatistics, University of South Carolina, Columbia, SC 29208, USA; 7Department of Nutrition, Connecting Health Innovations LLC, Columbia, SC 29201, USA

**Keywords:** airway inflammation, asthma, dietary inflammatory index, eosinophil, exercise, nutrients

## Abstract

Research suggests exercise may reduce eosinophilic airway inflammation in adults with asthma. The Dietary Inflammatory Index (DII^®^) quantifies the inflammatory potential of the diet and has been associated with asthma outcomes. This study aimed to determine whether the DII of a meal consumed either before or after exercise influences exercise-induced changes in airway inflammation. A total of 56 adults with asthma were randomised to (1) 30–45 min moderate–vigorous exercise, or (2) a control group. Participants consumed self-selected meals, two hours pre- and two hours post-intervention. Energy-adjusted DII (E-DII^TM^) was determined for each meal, with meals then characterised as “anti-inflammatory” or “pro-inflammatory” according to median DII. Induced sputum cell counts were measured pre- and four hours post-intervention. Participants consuming an anti-inflammatory meal two hours post-exercise had a decrease in sputum %eosinophils (−0.5 (−2.0, 0.3)%) compared with participants who consumed a pro-inflammatory meal two hours post-exercise (0.5 (0, 3.0)%, *p* = 0.009). There was a positive correlation between the E-DII score of the post-exercise meal and change in sputum %eosinophils (*r_s_* = 0.478, *p* = 0.008). The E-DII score of the meal consumed two hours pre-exercise had no effect on sputum %eosinophils (*p* = 0.523). This study suggests an anti-inflammatory meal two hours post-exercise augments exercise-induced improvements in eosinophilic airway inflammation in adults with asthma.

## 1. Introduction

Asthma is a heterogeneous disease of the airways usually characterised by chronic airway inflammation, and defined by a history of respiratory symptoms including wheeze, chest tightness, shortness of breath and cough [1]. There is some evidence that modifiable lifestyle factors, such as dietary intake and exercise, influence clinical asthma outcomes [2,3,4,5]. Because asthma is an inflammatory disease of the airways, it has been hypothesised that dietary intake and exercise may independently influence asthma outcomes via their effects on inflammation.

The general health benefits of exercise are believed to result, in part, from the modification of underlying chronic systemic and tissue-specific inflammation. There is some evidence that these anti-inflammatory effects may extend to the airways of adults with asthma. Exercise training has been shown to reduce eosinophilic airway inflammation and improve asthma-related quality of life in adults with asthma [3,6,7]. A single bout of exercise has been associated with a reduction in fractional exhaled nitric oxide (FeNO), a surrogate marker of eosinophilic airway inflammation [2]. A murine model of allergic asthma has also demonstrated a reduction in eosinophils and neutrophils in bronchoalveolar lavage, after a single bout of aerobic exercise [8]. Results from these studies suggest that both an acute bout of exercise and exercise training may reduce eosinophilic airway inflammation in people with asthma.

Some studies have shown that dietary intake also modulates inflammation in the airways of adults with asthma. The Dietary Inflammatory Index (DII^®^) is a tool that can be used to quantify the inflammatory potential of the diet or specific meals [9], and it is possible this may be a driving factor in modifying inflammation in the airways of people with asthma. A higher DII score is indicative of a more pro-inflammatory diet, which is typically higher in pro-inflammatory components such as saturated fat, and lower in anti-inflammatory components such as fibre and antioxidant nutrients such as flavonoids and certain vitamins [9]. The DII has been shown to be associated with mortality, cancer risk, cardiovascular disease and obesity [10]. The DII has also been shown to be associated with an increased risk of asthma, poorer lung function (FEV_1_) and increased systemic inflammation (IL-6) [11]. While research examining the impact of the DII in people with asthma is limited, studies have been conducted that examine the impact of various nutrients that comprise the DII on asthma outcomes. For example, a diet high in fruits and vegetables is negatively correlated with asthma prevalence [12,13] and, in people with asthma, has been shown to halve the risk of experiencing an asthma exacerbation [5]. Soluble fibre, which is found in fruits and vegetables as well as other plant foods including oats and barley, has been shown to reduce airway inflammation (FeNO, sputum neutrophils) in adults with asthma [14]. Lycopene, an antioxidant found predominately in tomatoes, has also been shown to reduce neutrophilic airway inflammation in adults with asthma [15]. Conversely, saturated fat has been shown to increase neutrophilic airway inflammation via activation of the nucleotide oligomerisation domain-like receptor protein 3 (NLRP3) inflammasome [4].

It is evident that both dietary factors and acute exercise independently modulate airway inflammation in people with asthma. However, whether the inflammatory potential of foods consumed either before or after exercise influences post-exercise changes in airway inflammation is not known. Therefore, the aim of this study was to determine the impact of the inflammatory potential of a meal consumed two hours pre-exercise, and two hours post-exercise, on changes in airway inflammation in adults with asthma.

## 2. Materials and Methods

### 2.1. Participants

Adults with asthma aged 18–55 years were recruited at the Princess Alexandra Hospital (Woolloongabba, Queensland, Australia) from respiratory outpatient databases and by advertisement, between April 2015 and June 2016. This is a secondary analysis of a randomised controlled trial [16]. Participants had physician diagnosed asthma with current episodic symptoms. They were classified as stable with no asthma exacerbation, respiratory tract infection, or oral corticosteroid use within the prior four weeks, and no hospitalisation for asthma in the three months prior to randomisation. Participants were non-smokers, either never smokers or having quit at least six months prior, with a BMI ≤ 40 kg/m^2^, an FEV_1_ ≥ 50% and currently undertaking ≤90 min structured moderate–vigorous intensity aerobic exercise per week. Exclusion criteria included participation in exercise of a moderate–vigorous intensity in the two days prior to the scheduled study visit; a cardiac, respiratory or musculoskeletal contraindication to exercise; uncontrolled hypertension; active cancer; diabetes mellitus; thyroid disorder; pregnancy or breastfeeding; and use of cholesterol-lowering or β-blocker medication. This project was approved by the Metro South Human Research Ethics Committee (HREC/15/QPAH/16) and all participants provided written informed consent. The trial was registered with the Australian New Zealand Clinical Trials Registry (http://www.anzctr.org.au; registration number ACTRN12615000294550) (accessed on 17 October 2022).

### 2.2. Experimental Protocol

The experimental protocol has been described in detail previously [16]. This was a parallel, randomised controlled trial in which participants were randomised to: (1) 30–45 min of moderate–vigorous intensity exercise (55–85% of predicted heart rate maximum, HRmax (220-participants age)), or (2) 30 min of rest in a chair (control group). All exercise was performed on a cycle ergometer (Corival, Lode BV, Groningen, The Netherlands). The pre-intervention meal (breakfast) was consumed two hours pre-intervention, while the post-intervention meal (lunch) was consumed two hours post-intervention (Appendix A).

Prior to attending the clinic, participants were advised to withhold bronchodilators and inhaled corticosteroid/bronchodilator combination medications for 6–24 h, and antihistamine medications for five days. Participants were asked to refrain from undertaking moderate–vigorous exercise within the two days prior to the study visit. A baseline induced sputum sample was collected to measure airway inflammation and spirometry was performed (Medgraphics, St Paul, MN, USA). Participants returned to the clinic the following morning. They were instructed to abstain from alcohol and caffeine in the 12 and 6 h prior to attending the clinic, respectively, and to consume breakfast 2 h prior. Salbutamol was provided (400 μg), then participants were randomised and completed their allocated intervention. Four hours post-intervention, a second induced sputum sample was collected (Appendix A). Randomisation codes were computer-generated random sequences, placed in sealed envelopes by an independent researcher. The exercise group completed a single session of either moderate or vigorous intensity exercise. After a five-minute warm-up (cycle load 7 watts), participants cycled for 40 min at 55–70% HRmax (moderate-intensity exercise) or for 25 min at 70–85% HRmax (vigorous-intensity exercise) followed by a one-minute cool down. The duration of the exercise differed so participants were matched according to kilojoules expended. Oxygen saturation, heart rate, dyspnoea and rating of perceived exertion [17] were measured at four minute intervals throughout the exercise intervention. The control group were instructed to remain seated for 30 min.

### 2.3. Participant Characterisation

Asthma control was measured by the Asthma Control Questionnaire (ACQ) [18] and asthma-related quality of life by the Asthma Quality of Life Questionnaire (AQLQ) [19]. Height and weight were measured with participants wearing no shoes and in light clothing. Waist circumference was measured in duplicate at the midpoint of the lowest rib and iliac crest (Lufkin W606PM Executive Diameter Tape, Lufkin, TX, USA) [20]. Atopy was determined by skin allergy testing. Atopy was defined as a wheal measuring ≥3 mm × 3 mm to any of the allergens measured. Allergens tested included house dust mite; four species of grass; cat hair; Aspergillus fumigatus; and Alternaria tenuis; as well as a positive (histamine) and negative control to ensure accuracy. Current exercise volume was determined by the International Physical Activity Questionnaire (IPAQ)—long form, with results expressed as metabolic equivalent tasks (METS) [21].

### 2.4. Calculation of the E-DII of Meals

Participants consumed a self-selected meal (breakfast) two hours prior to attending the clinic, which was recorded by the participant in a semi-quantitative food diary. Participants consumed a second self-selected meal two hours post-intervention (lunch), either from home or selected from the hospital cafeteria menu, which was recorded by the research dietitian. These meals were entered into a FoodWorks 8 (FoodWorks Professional Edition Version 8.0, Xyris Software Pty Ltd., Brisbane, Australia) database and nutrient analysis was performed, then the energy-adjusted DII (E-DII^TM^) was calculated from this nutrient analysis [9]. The E-DII is based on up to 45 food parameters including all macronutrients, energy, select micronutrients and antioxidants, cholesterol, fibre, caffeine and several herbs, spices and teas [22]. A total of 39 of these food parameters were available for inclusion in the current analysis (Appendix A). The E-DII score was computed to account for differences in energy intake [23]. A higher score was indicative of a more pro-inflammatory meal, while a lower score was indicative of a more anti-inflammatory meal [22]. For each meal, E-DII scores were divided into two groups: “anti-inflammatory” and “pro-inflammatory”, with the groups divided according to the median E-DII score of the cohort for each meal. Baseline E-DII was quantified using a 24-h semi-quantitative food diary, which was completed by participants the day prior to randomisation.

### 2.5. Sputum Inflammatory Markers

Sputum was induced over 15.5 min nebuliser time with 4.5% hypertonic saline [24]. Sputum was selected and dispersed using dithiothreitol. Total cell counts and cell viability (trypan blue exclusion) were determined. The sputum underwent differential cell count, which was determined from cytospins that had been prepared, stained (May-Grunwald-Giemsa) and counted from 400 non-squamous cells.

### 2.6. Statistics

Data were analysed using Stata 11.2 (Stata Corporation, College Station, TX, USA). Data are reported as mean ± standard deviation (SD) for parametric data or median (interquartile range (IQR)) for non-parametric data. Differences in the change in airway inflammation were examined according to intervention group and further divided according to whether the meal consumed was classified as anti-inflammatory or pro-inflammatory. The Kruskal–Wallis test with post-hoc Wilcoxon rank sum testing was used for group comparisons of airway inflammation, as these data were non-parametric. Correlations between pre- and post-intervention meal E-DII score and post-intervention nutrient intake with changes in eosinophilic airway inflammation were assessed using the Spearman rank test (*r_s_*). *p*-values < 0.05 were considered statistically significant. In analyses with four groups, a corrected *p*-value < 0.017 was used to account for multiple comparisons.

## 3. Results

### 3.1. Participant Demographics

Participants were recruited and completed the study between April 2015 and June 2016. A total of 60 participants were screened, and 56 were randomised and completed either a single bout of moderate–vigorous aerobic exercise (n = 38) or the control condition (n = 18) (Figure 1). Both groups were of a similar age, sex and body mass index (BMI) (Table 1), with 14 participants (25%) classified as obese. They also had similar lung function, asthma control (ACQ score), levels of airway inflammation, pack-year smoking history and baseline E-DII score (Table 1).

### 3.2. Impact of the Inflammatory Potential of a Meal Consumed Two Hours Post-Exercise

The median E-DII score of the meal consumed two hours post-intervention was −0.447 (range −3.702 to 3.869). Therefore, for the post-intervention meal analysis an anti-inflammatory meal was defined as a meal with an E-DII between −3.702 and <−0.447, while a pro-inflammatory meal was defined as a meal with an E-DII between −0.447 and 3.869.

In the exercise group, participants who consumed an anti-inflammatory meal two hours post-intervention had a significant decrease in sputum %eosinophils (−0.5 (−2.0, 0.3)%) compared with participants who consumed a pro-inflammatory meal two hours post-intervention (0.5 (0.0, 3.0)%, *p* = 0.009) (Figure 2, Table 2). Conversely in the control group, the post-intervention meal E-DII had no effect on the change in sputum eosinophils (*p* = 0.827) (Figure 2, Table 2). The combination of an anti-inflammatory meal with exercise (−0.5 (−2.0, 0.3)%) reduced sputum eosinophils compared with an anti-inflammatory meal without exercise (0.5 (0.0, 2.5)%, *p* = 0.009) (Figure 2, Table 2). In the exercise group, there was also a trend towards a reduction in sputum eosinophil count in participants who consumed an anti-inflammatory meal (−26 (−89, −4) × 10^4^/mL) compared to those who consumed a pro-inflammatory meal (5 (−1, 40) × 10^4^/mL); however, this was not statistically significant after adjustment for multiple comparisons (*p* = 0.029) (Table 2). The E-DII score of the post-intervention meal had no influence on sputum neutrophils, macrophages or lymphocytes (Table 2).

### 3.3. Impact of the Inflammatory Potential of a Meal Consumed Two Hours Pre-Exercise

The median E-DII score of the meal consumed two hours pre-intervention was 0.212 (range −3.012 to 4.298). Therefore, for the pre-intervention meal analysis an anti-inflammatory meal was defined as a meal with an E-DII between −3.012 and <0.212, while a pro-inflammatory meal was defined as a meal with an E-DII between 0.212 and 4.298. The E-DII of the meal consumed two hours pre-intervention did not correlate with the E-DII of the meal consumed two hours post-intervention (*r_s_* = −0.056, *p* = 0.702).

In the exercise group, the inflammatory potential of the meal consumed before exercise was not associated with the change in airway inflammation four hours post-exercise (Table 3). Participants in the control group who consumed an anti-inflammatory meal two hours pre-intervention had a significant increase in sputum %macrophages compared with participants in the exercise group who consumed a pro-inflammatory meal two hours pre-intervention (Table 3). There were no other effects of pre-intervention meal E-DII score on any markers of airway inflammation, in either the exercise or the control group (Table 3).

### 3.4. Correlations between Meal E-DII and Changes in Eosinophilic Airway Inflammation

In the exercise group, there was a positive correlation between the E-DII of the meal consumed two hours post-exercise and the change in both sputum %eosinophils (*r_s_* = 0.478, *p* = 0.008) and sputum eosinophil count (*r_s_* = 0.425, *p* = 0.030) (Figure 3). However, there was no correlation between the E-DII of the meal consumed two hours before exercise and change in sputum %eosinophils (*r_s_* = −0.072, *p* = 0.722) or change in sputum eosinophil count (*r_s_* = 0.155, *p* = 0.480) in the exercise group.

In the control group, there was no correlation between the E-DII score of the meal consumed two hours post-exercise and change in sputum %eosinophils (*r_s_* = 0.155, *p* = 0.480) or change in sputum eosinophil count (*r_s_* = 0.157, *p* = 0.576). There was a trend towards a positive correlation between the E-DII of the meal consumed two hours before exercise and change in sputum %eosinophils (*r_s_* = 0.473, *p* = 0.075); however, there was no correlation between pre-exercise meal DII and sputum eosinophil count (*r_s_* = 0.071, *p* = 0.817).

### 3.5. Effect of Nutrient Intake Two Hours Post-Intervention on Change in Eosinophilic Airway Inflammation

Regarding intake of specific nutrients at the meal two hours post-intervention, in the control group a greater intake of sugar (*r_s_* = 0.675, *p* = 0.006) and vitamin B6 (*r_s_* = 0.636, *p* = 0.011) correlated with a greater increase in sputum eosinophil count (Appendix A). Conversely, a greater intake of vitamin C two hours post-intervention correlated with a greater reduction in sputum eosinophil count (*r_s_* = −0.518, *p* = 0.048) (Appendix A).

In the exercise group, there was a trend towards a greater intake of long-chain omega-3 fatty acids (*r_s_* = −0.294, *p* = 0.145) and a lower intake of sodium (*r_s_* = 0.314, *p* = 0.118) being associated with a greater reduction in sputum eosinophil count; however, this did not reach statistical significance (Appendix A).

## 4. Discussion

This is the first study to demonstrate that an anti-inflammatory meal consumed two hours post-exercise augments exercise-induced improvements in airway inflammation at four hours in adults with asthma, with consumption of a more anti-inflammatory meal being associated with a greater decrease in sputum %eosinophils. Conversely, there was no association between the inflammatory potential of the meal consumed two hours before exercise and change in airway inflammation.

Our study demonstrates that consumption of an anti-inflammatory meal two hours post-exercise is associated with a greater post-exercise reduction in sputum eosinophils at four hours. Han et al. found that a higher DII score is associated with wheeze in adults, and that this effect is greatest in those with eosinophilic airway inflammation [25]. This, along with our study, suggests a relationship between the DII score and eosinophilic airway inflammation. A higher DII diet is typically higher in saturated fat and lower in fruits, vegetables, omega-3 fatty acids and fibre [11]. A recent study has demonstrated that children prescribed to follow the Mediterranean diet for six months had a significant reduction in the DII score of their diet, which was associated with a significant reduction in FeNO, an indirect measure of eosinophilic airway inflammation [26]. It has also been shown that a higher intake of fruits and vegetables is associated with a reduction in airway inflammation and improved lung function [27]. Antioxidants are predominantly found in fruits and vegetables. The absence of antioxidant consumption promotes circulating reactive oxygen species, which further promotes eosinophilic airway inflammation [28,29]. Our study did not highlight any one nutrient that was driving greater exercise-induced improvements in eosinophilic airway inflammation, suggesting it is the sum of the components of the E-DII rather than any one nutrient that is driving the anti-inflammatory effects we observed.

Exercise, both acute and prolonged, has previously been shown to reduce airway eosinophils in adults with asthma [2,3,8]. In our study, we found the decrease in eosinophils only appears to be apparent in participants who consumed an anti-inflammatory meal post-exercise; a combination previous research has not considered. In our previous study, an acute bout of moderate exercise was associated with a reduction in FeNO four hours post-exercise [2]. Participants fasted for the duration of our previous study; therefore, the exercise-induced anti-inflammatory effects were independent of dietary intake. Toennesen et al. found that a combined diet and exercise intervention over eight weeks significantly improved asthma control and quality of life, however there was no significant change in inflammatory biomarkers [30]. The authors attributed the results to a reduction in weight, which has previously been shown to improve asthma-related quality of life [3,31]. As our study featured acute exercise, weight loss was not a factor to consider for our participants.

We found no association between the E-DII of the meal consumed in the two hours prior to the intervention and airway inflammation. This suggests that it is the inflammatory potential of the meal consumed post-exercise, rather than before exercise, which is influencing changes in airway inflammation in our cohort. We also found no effect of the E-DII score of either meal on changes in airway inflammation in the control group. However, a greater intake of sugar and vitamin B6, and a lower intake of vitamin C, in the meal two hours post-intervention were associated with a greater increase in eosinophilic airway inflammation in the absence of exercise. A meta-analysis found vitamin C intake has a small beneficial effect on lung function (FEV_1_) [32], while a study conducted in children with asthma found vitamin C deficiency is associated with increased eosinophilic airway inflammation and asthma severity [33]. Regarding sugar, an isocaloric high-sucrose diet was associated with increased eosinophilic airway inflammation and increased airway resistance compared with a standard diet in a mouse model of allergic airway inflammation [33]. The association between vitamin B6 and sputum eosinophils is unclear and should be investigated in future studies.

This was the first study to examine the combined impact of the inflammatory potential of a meal and acute exercise on airway inflammation in adults with asthma. However, there are several limitations that must be acknowledged. This study included a relatively small sample size, particularly for the control group. However, the study included a well characterised cohort of participants and measured objective outcomes, increasing the strength of the findings. Another limitation was the completion of a semi-quantitative food diary pre-intervention, which may not have been an accurate representation of foods consumed. However, this type of food diary provided a high response rate from participants as it proved highly practical. The lunch meal following the intervention was recorded by a dietitian experienced in measuring dietary intake. Another limitation is that this study cannot prove causality, however it will inform the design of future studies that will be able to specifically address this issue. Additionally, participants in this study generally had well controlled asthma. Effect sizes may be larger in people with worse asthma control; this should be investigated in future studies. Finally, we cannot establish whether participation in the study influenced food choices, given participants knew dietary intake data was being recorded. However, we were interested in the effect of E-DII and specific nutrients on objectively measured changes in airway inflammation, which was not influenced by food choices.

## 5. Conclusions

This study demonstrates that consumption of an anti-inflammatory meal two hours post-exercise augments the anti-inflammatory effects of exercise in the airways of adults with asthma. Conversely, the inflammatory potential of the meal consumed two hours prior to exercise had no influence on airway inflammation. The findings of this study therefore suggest that consumption of an anti-inflammatory meal after, but not before, exercise may drive greater exercise-induced reductions in eosinophilic airway inflammation in adults with asthma. This has implications for asthma management, as it suggests consumption of an anti-inflammatory meal post-exercise may increase the beneficial effects of exercise in people with asthma. Further studies examining the impact of a post-exercise meal on asthma outcomes are warranted.

## Figures and Tables

**Figure 1 nutrients-14-04392-f001:**
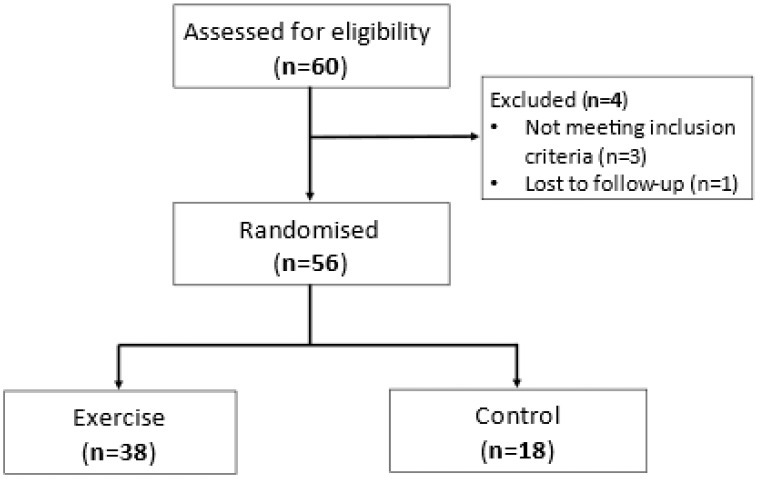
Flowchart of study participants.

**Figure 2 nutrients-14-04392-f002:**
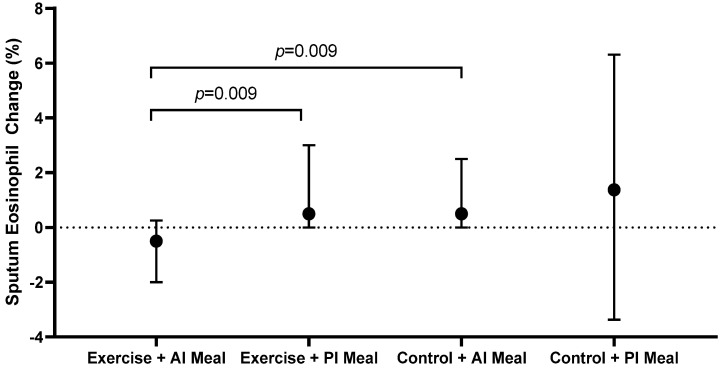
Change in sputum %eosinophils, according to intervention and inflammatory potential of the meal consumed two hours post-intervention. AI: anti-inflammatory; PI: pro-inflammatory.

**Figure 3 nutrients-14-04392-f003:**
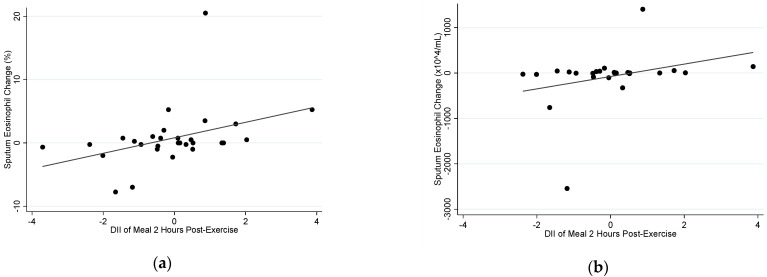
Correlation between the E-DII of the meal consumed two hours post-exercise with (**a**) sputum %eosinophil change at four hours (*r_s_* = 0.478, *p* = 0.008), and (**b**) sputum eosinophil count change at four hours (*r_s_* = 0.425, *p* = 0.030), following an acute bout of moderate–vigorous aerobic exercise.

**Table 1 nutrients-14-04392-t001:** Participant characteristics.

	Exercise	Control
N	38	18
Sex (n, %female)	26 (68.4)	16 (88.9)
Age (years)	32.8 (8.9)	34.8 (11.8)
BMI (kg/m^2^)	26.6 (4.7)	27.3 (5.4)
Waist circumference (cm)	88.7 (13.9)	86.8 (14.3)
Dietary Inflammatory Index		
Baseline	−0.11 (2.19)	−0.04 (1.94)
2 h pre-intervention	0.36 (1.70)	0.22 (1.40)
2 h post-intervention	−0.17 (1.40)	−0.75 (1.49)
Pre-BD FEV_1_ (% predicted)	87.4 (17.0)	81.6 (13.2)
Pre-BD FVC (% predicted)	93.3 (10.6)	91.3 (11.1)
Pre-BD FEV_1_/FVC (%)	77.2 (11.0)	74.4 (10.8)
Post-BD FEV_1_ (% predicted)	95.3 (13.3)	91.0 (10.6)
Post-BD FVC (% predicted)	96.4 (9.7)	95.5 (11.9)
Post-BD FEV_1_/FVC (%)	81.8 (8.7)	79.6 (8.8)
Atopy (n, %)	30 (78.9)	15 (83.3)
Former smokers (n, %)	6 (15.8)	2 (11.1)
Pack-years of former smokers *	2.6 (0.5, 6.7)	8.5 (6.8, 10.1)
Age at asthma diagnosis (years) *	5 (2, 11)	5 (3, 20)
ACQ score *	0.57 (0.29, 1.00)	0.64 (0.29, 1.00)
AQLQ score *	6.5 (5.8, 6.7)	6.4 (5.3, 6.8)
Total PA (METS) *	2631 (798, 7977)	3469 (2106, 7178)
Vigorous PA (METS) *	0 (0, 360)	43 (0, 401)
Moderate PA (METS) *	1080 (180, 5040)	1140 (720, 3535)
Walking (METS) *	677 (347, 2376)	1461 (693, 3432)
Airway Inflammation		
Total cell count (×10^6^/mL)	2.70 (1.62, 4.05)	2.34 (1.35, 4.14)
Eosinophils (%)	0.8 (0.0, 4.0)	1.1 (0, 2.5)
Eosinophils (×10^4^/mL)	19 (0, 186)	20 (6, 132)
Neutrophils (%)	14.9 (5.0, 28.0)	22.0 (4.0, 37.3)
Neutrophils (×10^4^/mL)	356 (144, 1389)	368 (97, 1780)
Macrophages (%)	66.6 (52.2, 78.8)	61.9 (46.8, 77.0)
Macrophages (×10^4^/mL)	1756 (1296, 2160)	1465 (827, 2071)
Lymphocytes (%)	2.0 (1.0, 4.8)	2.0 (0.5, 2.8)
Lymphocytes (×10^4^/mL)	62 (14, 115)	49 (17, 94)

Data presented as mean (SD), median (IQR) *, or n (%) the percentage of subjects with the specified variable. ACQ: Asthma Control Questionnaire; AQLQ: Asthma Quality of Life Questionnaire; BD: bronchodilator; BMI: body mass index; FEV_1_: forced expiratory volume in 1 s; FVC: forced vital capacity; IPAQ: International Physical Activity Questionnaire; METS: metabolic equivalent tasks (minutes per week); PA: physical activity.

**Table 2 nutrients-14-04392-t002:** Change in airway inflammation, according to intervention and inflammatory potential of the meal consumed two hours post-exercise.

	Exercise		Control		*p*-Value (Group Effect)
	Anti-Inflammatory Meal	Pro-Inflammatory Meal	Anti-Inflammatory Meal	Pro-Inflammatory Meal	
N	11	19	11	6	
Eosinophils (%)	−0.5 (−2.0, 0.3)	0.5 (0.0, 3.0) *	0.5 (0.0, 2.5) *	1.4 (−1.0, 2.5)	0.044
Eosinophils (×10^4^/mL)	−26 (−89, −4)	5 (−1, 40)	8 (4, 22)	78 (−24, 510)	0.049
Neutrophils (%)	7.3 (−1.3, 15.8)	8.3 (−3.5, 29.0)	21.3 (11.0, 35.5)	−1.3 (−10.5, 6.5)	0.115
Neutrophil (×10^4^/mL)	375 (71, 903)	320 (−9, 527)	256 (78, 742)	63 (−843, 857)	0.919
Macrophages (%)	−4.0 (−15.0, 6.0)	−5.3 (−25.0, 6.0)	−20.8 (−33.0, −2.5)	−0.8 (−5.8, 35.0)	0.284
Macrophages (×10^4^/mL)	−508 (−1263, 40)	−315 (−821, 5)	2 (−1017, 405)	408 (−832, 909)	0.607
Lymphocytes (%)	−1.0 (−4.0, 2.0)	0.0 (−1.8, 0.8)	−0.3 (−1.5, 0.8)	0.5 (−0.3, 2.5)	0.702
Lymphocytes (×10^4^/mL)	19 (−58, 73)	8 (−15, 60)	4 (−81, 21)	23 (−16, 52)	0.884

Data presented as median (IQR). * *p* < 0.017 versus the exercise + anti-inflammatory meal group.

**Table 3 nutrients-14-04392-t003:** Change in airway inflammation, according to intervention and inflammatory potential of the meal consumed two hours pre-exercise.

	Exercise		Control		*p*-Value (Group Effect)
	Anti-Inflammatory Meal	Pro-Inflammatory Meal	Anti-Inflammatory Meal	Pro-Inflammatory Meal	
N	12	15	8	7	
Eosinophils (%)	0.3 (−1.3, 3.3)	0.0 (−1.0, 0.8)	0.3 (−10.5, 1.6)	1.3 (0.3, 6.5)	0.184
Eosinophils (×10^4^/mL)	−2 (−89, 54)	5 (−12, 32)	15 (3, 142)	9 (4, 483)	0.320
Neutrophils (%)	7.0 (−1.8, 10.6)	10.0 (−2.5, 30.5)	9.9 (−6.3, 28.0)	19.3 (8.0, 35.5)	0.259
Neutrophil (×10^4^/mL)	251 (71, 457)	320 (−9, 903)	222 (146, 742)	78 (−843, 918)	0.909
Macrophages (%)	−4.3 (−11.0, 3.1)	−4.0 (−24.8, 7.0)	−7.9 (−25.8, 6.3)	−20.8 (−24.0, −5.5)	0.650
Macrophages (×10^4^/mL)	−129 (−502, 88)	−315 (−601, −70)	413 (109, 641) *	−1017 (−2195, 885)	0.049
Lymphocytes (%)	0.0 (−1.9, 0.6)	0.0 (−2.6, 1.8)	0.4 (−1.5, 1.0)	−0.3 (−0.5, −0.3)	0.717
Lymphocytes (×10^4^/mL)	9 (−16, 60)	8 (−58, 61)	31 (−16, 63)	−81 (−128, 21)	0.247

Data presented as median (IQR). * *p* < 0.017 versus the exercise + pro-inflammatory meal group.

## Data Availability

The data presented in this study are available on request from the corresponding author.

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
