# Peer review of "The Impact of Meal Dietary Inflammatory Index on Exercise-Induced Changes in Airway Inflammation in Adults with Asthma"

_nutrients, 2022, doi:10.3390/nu14204392_

Round 1
Reviewer 1 Report
This intervention study examined the impact of Meal dietary inflammatory index on exercise-induced changes in airway inflammation in adults with asthma. The study reported that an anti-inflammatory meal two-hours post-exercise reduced exercise-induced improvement in eosinophilic airway inflammation in adults with asthma. This study highlights the importance of diet of anti-inflammatory properties on asthma management. However, there are few comments and questions:
1) One of the major concerns of the study is that there was no baseline E-DII of the participants. It is important to show that there was no differences of the dietary quality between exercise and control group when randomized
2) Alternatively, a case-crossover design can be considered. The comparisons are made within individuals that can eliminate several confounding effects
3) In addition to small sample size, the numbers between the group were not balanced. Please explain why there are as twice as much in exercise group (n=38) as comparing to control group (n=18)
4) Please show the median and IQR for BMI in Table 1. It seemed that the criteria of BMI <=40 kg/m2 was very wide. Please also indicate how many participants are overweight or obese.
5) Although it was stated that atopy was determined by skin allergy testing (line 141), please specify how atopy was defined. Similarly, please state how total PA (METS) was estimated.
Minor comments:
1) In Table S1, only 34 nutrients were shown. If so, please indicate that only 34 out of 45 food parameters were included in the study
2) In Table 1, please double check the IRQ for pack years of smoking. If it included former smokers, the range should not be 0
Author Response
The Impact of Meal Dietary Inflammatory Index on Exercise-Induced Changes in Airway Inflammation in Adults with Asthma
Author Responses to Reviewer 1:
This intervention study examined the impact of Meal dietary inflammatory index on exercise-induced changes in airway inflammation in adults with asthma. The study reported that an anti-inflammatory meal two-hours post-exercise reduced exercise-induced improvement in eosinophilic airway inflammation in adults with asthma. This study highlights the importance of diet of anti-inflammatory properties on asthma management.
Response:
Thank you for your review of our manuscript. We have provided a point-by-point response to your comments and questions below.
- One of the major concerns of the study is that there was no baseline E-DII of the participants. It is important to show that there was no differences of the dietary quality between exercise and control group when randomized.
- Alternatively, a case-crossover design can be considered. The comparisons are made within individuals that can eliminate several confounding effects
Response:
Baseline E-DII was quantified using a semi-quantitative 24-hour food diary, which was completed by participants the day prior to randomisation. This food diary determined there was no difference in baseline E-DII between the exercise versus control group (-0.11±2.19 vs -0.04±1.94, p=0.910). These data have now been added to Table 1 and are described in text (line 193-195):
“They also had similar lung function, asthma control (ACQ score), levels of airway inflammation, pack-year smoking history and baseline E-DII score (Table 1).”
The method for determining baseline E-DII has been added to the Methods section (line 165-166):
“Baseline E-DII was quantified using a 24-hour semi-quantitative food diary, which was completed by participants the day prior to randomisation.”
- In addition to small sample size, the numbers between the group were not balanced. Please explain why there are as twice as much in exercise group (n=38) as comparing to control group (n=18)
Response:
This is a secondary analysis of a study in which participants were randomised to complete either moderate exercise, vigorous exercise, or the control intervention (DOI: https://doi.org/10.1513/AnnalsATS.202109-1053OC). The current manuscript aims to determine whether the inflammatory potential of the meal consumed either pre- or post-intervention influences changes in airway inflammation following exercise in general. We did not observe any effect of exercise intensity on this response. As such, the moderate and vigorous groups were combined, resulting in a larger exercise group.
- Please show the median and IQR for BMI in Table 1. It seemed that the criteria of BMI <=40 kg/m2 was very wide. Please also indicate how many participants are overweight or obese.
Response:
The median (IQR) BMI was 27.7 (22.3, 30.3)kg/m2 in the exercise group and 26.9 (23.2, 29.9)kg/m2 in the control group. As the BMI data are normally distributed, results have been presented as mean (SD) as per the statistical analysis plan (line 176-177), which also provides information on the spread of the data around the mean.
Fourteen participants (25%) were obese, while 42 (75%) were non-obese. This information has been added to the Results section (line 192-193):
“Both groups were of a similar age, sex and body mass index (BMI) (Table 1), with 14 participants (25%) classified as obese.”
- Although it was stated that atopy was determined by skin allergy testing (line 141), please specify how atopy was defined. Similarly, please state how total PA (METS) was estimated.
Response:
Detail regarding how atopy was defined and additional detail regarding how METS were estimated has been added to the manuscript (line 141-146; specific detail regarding calculation of METS from the IPAQ can be found in the paper cited in the manuscript):
“Atopy was defined as a wheal measuring ≥3mm x 3mm to any of the allergens measured. Allergens tested included house dust mite; four species of grass; cat hair; aspergillus fumigatus; and alternaria tenuis; as well as a positive (histamine) and negative control to ensure accuracy. Current exercise volume was determined by the International Physical Activity Questionnaire (IPAQ) – long form, with results expressed as metabolic equivalent tasks (METS) [23].”
Minor comments:
- In Table S1, only 34 nutrients were shown. If so, please indicate that only 34 out of 45 food parameters were included in the study
Response:
Thirty-nine nutrients were included in the calculation of the E-DII; these nutrients have now been listed in a new table (Table S1).
- In Table 1, please double check the IRQ for pack years of smoking. If it included former smokers, the range should not be 0
Response:
This data has been checked and is correct. The data presented is IQR, not range. Because only 14% of participants were former smokers, this represents only a small portion of participants and, thus, the overall pack years of this cohort is extremely low. To avoid confusion, the pack-year data have been updated and now only includes the 14% of participants who were former smokers (Table 1).
Reviewer 2 Report
Dear Editor,
The coauthors in the manuscript with title "The Impact of Meal Dietary Inflammatory Index on Exercise-Induced Changes in Airway Inflammation in Adults with Asthma" recruited 56 asthmatic adults and devided them into two groups. One group (N=38) performed physical excercise and the other group (N=18) did not and was considered as controls. Both groups were served food meals before and after excercise in a two-hour interval. The coauthors concluded that the meal was served two hours after excercise played an anti-inflammatory role against asthma through reducing the sputum eosinophil counts compared to control group.
The manuscript is very interesting. However I would recommend the coauthors to create a flow chart for the whole study setup, especially for the meals were served, because it is too much information at once.
Author Response
The Impact of Meal Dietary Inflammatory Index on Exercise-Induced Changes in Airway Inflammation in Adults with Asthma
Author Responses to Reviewer 2:
The coauthors in the manuscript with title "The Impact of Meal Dietary Inflammatory Index on Exercise-Induced Changes in Airway Inflammation in Adults with Asthma" recruited 56 asthmatic adults and devided them into two groups. One group (N=38) performed physical excercise and the other group (N=18) did not and was considered as controls. Both groups were served food meals before and after excercise in a two-hour interval. The coauthors concluded that the meal was served two hours after exercise played an anti-inflammatory role against asthma through reducing the sputum eosinophil counts compared to the control group.
The manuscript is very interesting. However I would recommend the coauthors to create a flow chart for the whole study setup, especially for the meals were served, because it is too much information at once.
Response:
Thank you for your review of our manuscript and suggestion to add a flow-chart for the study set up. Please see additional figure (Figure S1), which we believe adds clarity to the paper.